# CAN DECODING BY CONTRASTING LAYERS REALLY IMPROVE FACTUALITY IN LARGE LANGUAGE MODELS?

## ABSTRACT

Large language models (LLMs) have made notable advancements across diverse applications, but their susceptibility to hallucinations remains a critical challenge. That is, they could produce outputs divergent from real-world evidence or user-provided inputs. Recent studies have explored a contrastive decoding strategy known as DoLa, which mitigates output inaccuracy by contrasting the outputs from the final layer against those from the previous layers. Nevertheless, such strategy has its limitation, as LLMs, which already have internalized extensive parametric knowledge through comprehensive pre-training and fine-tuning phases, may generate errors due to incorrect or obsolete information within their parameters. As an alternative, external knowledge could be included in the prompt context for querying, but the constrained context window of LLMs poses a significant barrier restricting the amount of information that can be provided.

To address the above issues, we propose to integrate the contrastive decoding strategy with a long-context encoder that effectively condenses extensive initial contexts into a more concise format. Additionally, our approach employs an adaptive decoding mechanism that dynamically selects between standard decoding and contrastive decoding based on the model's prediction uncertainty, quantified using entropy. Extensive experiments have demonstrated that our proposed methodology enhances the factual accuracy of the produced content when applied to various datasets. For instance, it has improved the performance of LLaMA2-7B models on the Quality dataset by 61.61%, compared to the DoLa decoding method, showcasing its effectiveness in enhancing the reliability of LLMs in generating truthful information.

## 1 INTRODUCTION

Large language models (LLMs) have demonstrated significant achievements across diverse tasks, yet they are prone to generating hallucinations—outputs that deviate from user inputs, conflict with earlier context, or are inconsistent with factual data. This propensity challenges their deployment in critical domains such as healthcare and finance, where accuracy and reliability are paramount. Numerous studies have sought to address these issues, employing strategies categorized into mitigation during training (Zhou et al., 2023; Penedo et al., 2023; Li et al., 2023c; Lee et al., 2023; Zhong et al., 2021a; Chen et al., 2024a; Cao et al., 2023; Lee et al., 2024), reinforcement from human feedback (Ouyang et al., 2022; Zheng et al., 2023), and inference (Manakul et al., 2023; Du et al., 2023; Liang et al., 2023; Li et al., 2023b). The recent introduction of the DoLa (Chuang et al., 2024) method, which utilizes contrastive decoding during inference, has made strides in reducing hallucinations by extracting and contrasting probability distributions from earlier and final model layers, thus emphasizing more reliable information.

However, the efficacy of DoLa relies on the substantial parametric knowledge embedded within LLMs during extensive pre-training and fine-tuning phases (Roberts et al., 2020), which can harbor inaccuracies leading

to hallucinations (Zhang et al., 2023). A potential solution is to integrate current, verified knowledge from trusted sources as a form of dynamic updating (Li et al., 2022a; Lewis et al., 2021; Borgeaud et al., 2022; Liu et al., 2023; Li et al., 2024). Nonetheless, the integration of such extensive external knowledge remains challenging due to the constrained context window size of LLMs, which restricts the amount of information that can be effectively processed without exceeding the model's context limits.

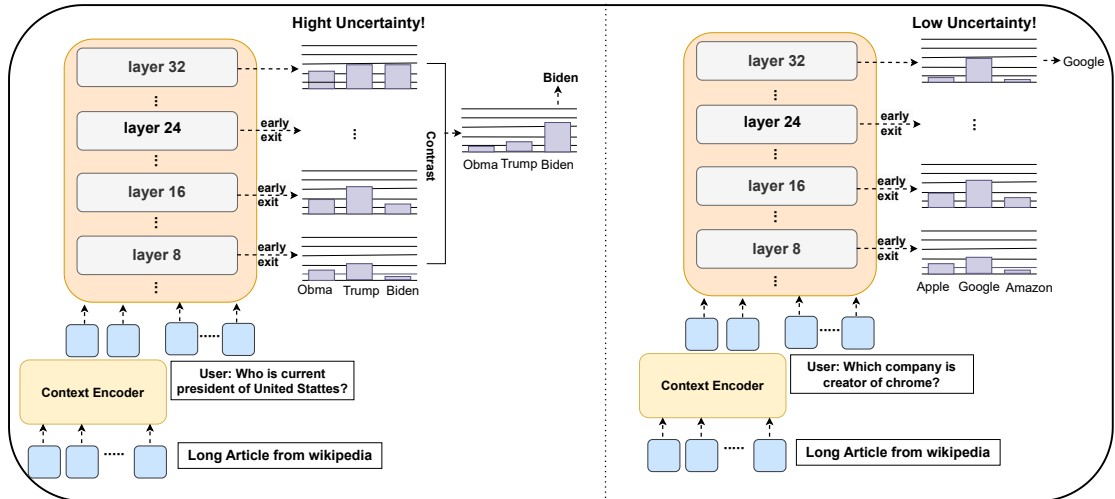

Figure 1: Contrast between the DoLa (left) and our proposed method (right) is elucidated here. The DoLa method involves contrasting probability distribution from the final layer against those from earlier layers, relying on the model's inherent parametric knowledge to derive outputs. Conversely, in our approach, the extensive context accompanying the user's prompt undergoes initial processing via a context encoder. The resultant embeddings from this process are then affixed to the embeddings from the user's prompt. These combined embeddings are subsequently fed into the LLM, which generates its outputs by contrasting information from the last layer against that from preceding layers.

To address this constraint, we have employed strategies derived from long-context LLMs and devised a method that incorporates a context encoder (Chevalier et al., 2023). This encoder is specifically designed to transform extensive, detailed contexts into considerably more concise representations. This compression facilitates the integration of external knowledge, which compensates for the limitations inherent in the DoLa method, particularly its reliance on potentially outdated or incorrect parametric knowledge. The resulting token embeddings, referred to as summary embeddings, are significantly more compact than the original contexts. These summary embeddings are then combined with the user's prompt to form the final prompt for the LLM. In response, the LLM generates outputs by contrasting the probability distributions across its lower and upper layers, with the process being influenced by both the summary embeddings and the user's initial prompt.

Building upon the integration of external knowledge, we introduce an adaptive decoding strategy that dynamically selects between standard decoding and contrastive decoding based on the uncertainty of the model's predictions. This uncertainty is quantified using the entropy of the logits from the final transformer layer. Specifically, when the entropy is low, indicating high confidence in the model's predictions, standard decoding is employed to generate responses. Conversely, when the entropy is high, indicating uncertainty, contrastive decoding is utilized to refine the predictions by leveraging the discrepancies between intermediate and final layer outputs. This adaptive approach ensures that the model maintains high accuracy and reliability, especially in scenarios where the likelihood of hallucinations is elevated.

To elucidate our concept more clearly, we illustrate our approach in Figure 1. Each figure demonstrates the integration of external knowledge. The left panel depicts a scenario with high entropy in the last layer logits, indicating high uncertainty, thereby prompting the use of contrastive decoding. In contrast, the right panel shows a scenario with low entropy in the last layer logits, indicating low uncertainty, and thus the model employs standard decoding. In both cases, external knowledge is effectively integrated through the context encoder, which condenses extensive external information into compact embeddings suitable for processing within the LLM's limited context window. This adaptive decoding strategy ensures accurate and reliable response generation tailored to the model's confidence level.

Extensive experimental evaluations demonstrate that our approach either surpasses or is comparable to existing baselines. Specifically, in the context of Quality dataset designed for a question-answering task, our method significantly outperforms DoLa. It achieves an accuracy score of 53.93%. Compared with the DoLa method employing contrastive decoding, our approach exhibits a more substantial advantage, enhancing the score by an additional 20.56%. Moreover, even when the DoLa method is augmented with truncated context, our method still demonstrates superior performance across various datasets. For instance, on the Qasper dataset, our method achieves an F1 score of 31.43%, whereas the DoLa method with provided context only reaches an F1 score of 17.41%. On average, our method attains a score of 29.29%. In contrast, the DoLa method records lower average scores of 18.67% without context and 24.49% with context. These results collectively validate that our method can significantly reduce hallucinations in large language models.

## 2 RELATED WORKS

**Contrastive Decoding.** Contrastive decoding was initially conceived to enhance the fluency and coherence of generation by large language models (LLMs) (Li et al., 2022b), by contrasting the output probabilities between expert-level LLMs and their less advanced counterparts. Building on this foundation, a subsequent study by (Shi et al., 2023) introduced context-aware decoding, designed to augment LLMs' focus on contextual nuances in summarization tasks and reduce the occurrence of knowledge discrepancies. More recently, the Autocomptrastive decoding method (Gera et al., 2023) was proposed to improve diversity and coherence in smaller models such as the GPT2 125M, primarily by fine-tuning the prediction head in early layers. Furthermore, DoLa (Chuang et al., 2024) was proposed to enhance factual accuracy and reduce hallucinations in LLMs, by dynamically selecting early layers based on the complexity of tokens, thereby circumventing extensive training costs. This approach has been successfully applied to larger models, from LLaMa 7b to LLaMa 70b, demonstrating notable efficacy.

**Long-context LLMs.** Various approaches have been explored to enhance an LLM's capacity of accepting long contexts. The Rotary Position Embeddings (RoPE) (Su et al., 2021) has enabled the handling of longer contexts, extending up to 128,000 tokens (Chen et al., 2023; 2024b; Peng et al., 2024). Mistral (Jiang et al., 2023) has introduced a sliding window attention mechanism that focuses only on a subset of tokens from the preceding layer. Another research trajectory aimed at creating a versatile compressor capable of condensing any input prompts. Examples include GIST (Mu et al., 2023), AutoCompressor (Chevalier et al., 2023), and ICAE (Ge et al., 2023). A recent study (Tan et al., 2024) implemented a parameter-efficient fine-tuning (PEFT) method, LoRa (Hu et al., 2021), to mitigate these issues. Despite its efficacy, this method remains computationally demanding. Consequently, we have opted to use another PEFT method, IA3 (Liu et al., 2022), which offers a more computationally efficient solution for aligning compressed embeddings with the original embeddings. This lighter fine-tuning approach enables us to more effectively incorporate extensive external knowledge into LLMs, thereby enhancing the fidelity of generated content.

## 3 METHODOLOGY

A language model comprises an initial embedding module, denoted as $E$, followed by $L$ consecutive transformer blocks labeled $T_1, T_2, \ldots, T_L$, and a final linear projection layer $\Psi(\cdot)$ that calculates the probability distribution for the next token. For an input token sequence $\{w_0, w_1, \ldots, w_{k-1}\}$, the inference process begins with the embedding module converting these tokens into a sequence of vectors $V_0 = \{v_0^{(0)}, \ldots, v_{k-1}^{(0)}\}$. This vector sequence $V_0$ is then iteratively refined through each transformer block as follows:

$$\Psi(T_L(T_{L-1}(\ldots T_1(E(w_0, w_1, \ldots, w_{k-1}))))).$$

Here, the output of the $m$-th transformer block is represented by $V_m = T_m(T_{m-1}(\ldots T_1(E(w_0, w_1, \ldots, w_{k-1}))))$. The final projection layer $\Psi(\cdot)$ computes the probability of the token $w_k$ within the vocabulary $\mathcal{V}$:

$$P(w_k \mid w_{<k}) = \text{softmax}\big(\Psi(v_k^{(L)})\big)_{w_k}, \quad w_k \in \mathcal{V}.$$

Unlike approaches that apply $\Psi$ exclusively at the final layer, the DoLa (Chuang et al., 2024) method leverages discrepancies between intermediate and higher transformer layers to ascertain the probability of the subsequent token. However, DoLa is constrained by its reliance on the model's intrinsic parametric knowledge and does not incorporate external knowledge repositories, limiting its ability to rectify inaccuracies embedded during initial training.

### 3.1 INCORPORATING EXTERNAL KNOWLEDGE

To overcome the limitations of relying solely on parametric knowledge, we integrate external knowledge through context-enhanced prompt-based querying. The finite size of an LLM's context window poses challenges for directly embedding extensive contextual information. To address this, we introduce a context compression mechanism that reduces the original, extensive contexts into a more compact form (Chevalier et al., 2023; Tan et al., 2024). Specifically, we employ a context encoder, denoted as $C$, which operates as a language model that accepts a sequence of tokens and produces a corresponding sequence of condensed token embeddings, referred to as summary embeddings. These summary embeddings are significantly shorter than the original contexts, facilitating their incorporation into the model without exceeding the context window limitations.

We utilize the AutoCompressor (Chevalier et al., 2023), a specialized context compression model fine-tuned for the LLaMA2-7B architecture, as our context encoder. AutoCompressor effectively segments lengthy contexts into chunks of 1536 tokens and recursively compresses each chunk into summary embeddings, each consisting of 50 tokens (Chevalier et al., 2023). These summary embeddings function as pseudo-words within the LLM decoder's embedding space, encapsulating high-level abstractions or summaries (Tan et al., 2024). Following the methodology outlined in (Tan et al., 2024), we initialize the LLaMA2-7B model with weights optimized by the AutoCompressor.

### 3.2 PARAMETER-EFFICIENT FINE-TUNING

To align the summary embeddings generated by the context encoder with the embedding space of the target LLM, we apply a Parameter-Efficient Fine-Tuning (PEFT) strategy on the training subset of our datasets. Specifically, we utilize the $(IA)^3$ technique (Liu et al., 2022), which scales activations using learned vectors, thereby enhancing performance with minimal additional parameters. Within each attention layer, we introduce and train three learned vectors: $\ell_{\text{key}} \in \mathbb{R}^{d_{\text{key}}}$, $\ell_{\text{val}} \in \mathbb{R}^{d_{\text{val}}}$, and $\ell_{\text{ff}} \in \mathbb{R}^{d_{\text{ff}}}$, corresponding to the key vector $K$, value vector $V$, and query vector $Q$, respectively. These vectors are integrated into the model's attention mechanisms as follows:

$$\text{softmax}\left(\frac{Q(\ell_{\text{key}} \odot K^\top)}{\sqrt{d_{\text{key}}}}\right)(\ell_{\text{val}} \odot V)$$

and within the feed-forward networks as $(\ell_{\text{ff}} \odot \gamma(w_0 x))W_2$, where $\gamma$ denotes the non-linear activation function in the feed-forward layer.

## 3.3 ADAPTIVE DECODING STRATEGY

To enhance the robustness and reliability of the decoding process, we introduce an adaptive decoding strategy that dynamically selects between standard decoding and contrastive decoding based on the uncertainty of the model's predictions. This uncertainty is quantified using the entropy of the logits from the final transformer layer.

**Entropy-Based Decision Making**    Entropy serves as a measure of uncertainty in the model's predictions. For the probability distribution $P(w_k \mid w_{<k})$ over the vocabulary $\mathcal{V}$, the entropy $H$ is defined as:

$$H(P) = -\sum_{w \in \mathcal{V}} P(w \mid w_{<k}) \log P(w \mid w_{<k}). \tag{1}$$

A low entropy value indicates high confidence in the prediction, as the probability mass is concentrated on a few tokens. Conversely, a high entropy value signifies greater uncertainty, with the probability distribution being more spread out.

Based on the entropy $H(P)$, we adaptively choose the decoding strategy:

$$\hat{P}(w_k \mid C(c), w_{<k}) = \begin{cases} P_{\text{normal}}(w_k \mid C(c), w_{<k}), & \text{if } H(P) \leq \tau, \\ P_{\text{contrast}}(w_k \mid C(c), w_{<k}), & \text{if } H(P) > \tau, \end{cases} \tag{2}$$

where $\tau$ is a predefined entropy threshold that determines whether the model is in a state of low or high uncertainty.

**Normal Decoding**    When the entropy $H(P)$ is below or equal to the threshold $\tau$, indicating low uncertainty, the model proceeds with standard decoding. In this scenario, the next token probability is directly obtained from the final projection layer:

$$P_{\text{normal}}(w_k \mid C(c), w_{<k}) = P(w_k \mid C(c), w_{<k}).$$

**Contrastive Decoding**    When the entropy $H(P)$ exceeds the threshold $\tau$, indicating high uncertainty, the model employs contrastive decoding as described in the DoLa method (Chuang et al., 2024). Specifically, the probability distribution is computed by contrasting the outputs of an intermediate layer with the final layer (Chuang et al., 2024). The selection of the intermediate layer $M$ is based on maximizing the divergence from the final layer $L$ using Jensen-Shannon (JS) divergence:

$$M = \arg\max_{j \in \mathcal{L}} \text{JS-Divergence}\big(Q_L(\cdot \mid C(c), w_{<k}) \parallel Q_j(\cdot \mid C(c), w_{<k})\big)$$

where $\mathcal{L}$ represents the pool of potential layers eligible for selection as the intermediate layer.

Following the selection, the difference is calculated by subtracting the logarithmic probabilities of the intermediate layer's output from those of the final layer (Li et al., 2022b; Chuang et al., 2024):

$$\hat{P}(w_k \mid C(c), w_{<k}) = \text{softmax}\big(\mathcal{G}\big(Q_L(C(c), w_k), Q_M(C(c), w_k)\big)\big)_{w_k},$$

$$\text{where} \quad \mathcal{G} = \begin{cases} \log \dfrac{Q_L(C(c), w_k)}{Q_M(C(c), w_k)}, & \text{if } w_k \in \mathcal{V}_{\text{head}}\,(w_k \mid w_{<k}), \\ -\infty, & \text{otherwise.} \end{cases}$$

The subset $\mathcal{V}_{\text{head}} \subset \mathcal{V}$ is determined based on whether a token attains sufficiently high probability from the final layer (Li et al., 2022b; Chuang et al., 2024):

$$\mathcal{V}_{\text{head}}\left(w_k \mid C(c), w_{<k}\right) = \left\{w_k \in \mathcal{V} \mid Q_L(C(c), w_k) \geq \alpha \max_w Q_L(C(c), w)\right\}.$$

This mechanism ensures that when the model is uncertain, it leverages the contrastive decoding approach to refine its predictions by emphasizing the divergence between intermediate and final layers, thereby enhancing the accuracy of high-uncertainty predictions.

**Entropy Threshold Selection**  The threshold $\tau$ is empirically determined based on validation performance to balance between standard and contrastive decoding. An appropriate $\tau$ ensures that the model switches to contrastive decoding only when necessary, maintaining efficiency by avoiding unnecessary contrastive computations during low-uncertainty scenarios.

**Repetition Penalty**  Consistent with the approach in (Keskar et al., 2019; Chuang et al., 2024), to mitigate issues related to repetitive text generation, we incorporate a simple repetition penalty during the decoding process. This penalty discourages the model from generating the same token repeatedly, thereby enhancing the diversity and coherence of the generated text.

**Final Probability Computation**  Integrating the adaptive decoding strategy, the final probability $\hat{P}(w_k \mid C(c), w_{<k})$ is computed as:

$$\hat{P}(w_k \mid C(c), w_{<k}) = \begin{cases} P_{\text{normal}}(w_k \mid C(c), w_{<k}), & \text{if } H(P) \leq \tau, \\ \text{softmax}\big(\mathcal{G}\big(Q_L(C(c), w_k), Q_M(C(c), w_k)\big)\big)_{w_k}, & \text{if } H(P) > \tau. \end{cases}$$

This adaptive approach ensures that the model dynamically adjusts its decoding strategy based on the confidence of its predictions, thereby optimizing both the accuracy and fluency of the generated text.

## 4 EXPERIMENTS

### 4.1 SETUP

To evaluate the efficacy of our method, we undertake an extensive series of experiments. The primary objectives of this empirical investigation are to determine: (1) whether the existing contrastive decoding method can mitigate hallucinations in the absence of parametric knowledge associated with the user input, (2) the capability of our method to perform effectively when managing extended contexts, and (3) the additional inference latency incurred by our method.

**Datasets.** Our investigation centers on the tasks of question answering and summarization. For question answering, we utilize four distinct datasets: **Quality** (Pang et al., 2021), **Qasper** (Dasigi et al., 2021), **NarrativeQA** (Kočiský et al., 2018) and **HotpotQA** (Yang et al., 2018). For summarization, we use the **QMSum** (Zhong et al., 2021b) dataset. Only the validation partitions of the above datasets are used in our evaluation. Note that these datasets contain samples with extensive textual contexts that provide necessary information to facilitate response generation for posed questions. Different evaluation metrics are tailored to the specific characteristics of each dataset. For the Quality dataset, we use the accuracy score as the evaluation metric. For QMSum, we employ the geometric mean of the ROUGE scores. For Qasper, NarrativeQA and HotpotQA, we utilize F1 scores. More details about these datasets can be found in the Appendix.

**Models and Baselines.** We utilize the Llama-2-7B model (Touvron et al., 2023) with a context window of 4096 tokens (denoted as Llama-2-7B-4k) as the foundational model, as it is renowned for its state-of-the-art performance across a variety of tasks and its broad applicability in diverse contexts. To condense extensive textual contexts, we employ AutoCompressor (Chevalier et al., 2023), a context compressor meticulously

fine-tuned for Llama-2-7B, which not only generates summary tokens from substantial contexts but also supports text completions derived from these tokens (Chevalier et al., 2023). For Llama-2-7B-4k, we initialize its weights optimized by AutoCompressor, and apply a Parameter Efficient Fine Tuning (PEFT) technique, specifically the IA$^3$ method (Liu et al., 2022), to further refine the alignment between its embedding space and the summary embeddings. The details can be found in in Huggingface [1].

For the purpose of comparison, we evaluate the following baselines (similar to (Tan et al., 2024)): (1) Llama-2-7B-4k (Touvron et al., 2023), with and without additional context. (2) Llama-2-7B-32k (Li et al., 2023a), a variant of the Llama-2 model fine-tuned to accommodate a more expansive context window of 32,000 tokens and enable position interpolation; with and without supplementary context. (3) Llama-2-7B-4k and Llama-2-7B-32k further fine-tunned to adapt the entire model to extensive context situations, specifically within the training partition of each dataset. (4) Llama-2-7B-4k and Llama-2-7B-32k used with the Retrieval mechanism, where each document is segmented into fragments of 512 tokens, the Contriever (Izacard et al., 2021) is utilized to extract the top five most pertinent segments, and the segments are subsequently merged with the original user input to be fed into the LLM. (5) Llama-2-7B-4k enhanced with DoLa, with and without provided context.

**Candidate Layers** To effectively contrast the probability distances between layers, we designate a spectrum of layers as candidates. Given the architecture of Llama-2-7B, which consists of 32 layers, we adopt the framework suggested in (Chuang et al., 2024) for layer selection. Specifically, we select candidate layers from 0 to 16 with a two-layer gap. For the DoLa baseline, the details can be found in Appendix. We also conduct ablation studies to evaluate the impact of selection strategy.

## 4.2 QUESTION ANSWERING TASK

We evaluated the efficacy of our method relative to various benchmarks, and our findings indicate that the performance of our method either surpasses or equates to that of the baselines.

**Qasper** Our method achieved the highest F1 score of 31.43%, marginally surpassing the top baseline performance of 29.71% attained by the LLaMa-2-7B-32k model with comprehensive fine-tuning. Notably, full fine-tuning incurs significant computational costs; in contrast, our method is more efficient, consuming considerably fewer resources in terms of time and computational expenses. Although DoLa outperforms the original decoding approach on LLaMa-2-7B, with F1 scores of 14.49% versus 7.68% in scenarios without context, our method substantially surpasses this performance. Specifically, DoLa without context achieved an F1 score of 14.49%, which is 16.94% lower than that of our method. Even when context is provided, DoLa achieves an F1 score of 17.42%, still falling short of our method's 31.43%.

**Narrative QA** For this dataset, the LLaMa-2-7B model with fine-tuning exhibited strong performance, achieving an F1 score of 28.72%. Nonetheless, our method outperformed DoLa both with and without context. Specifically, DoLa without context registered an F1 score of 12.80%, which is 8.75% lower than our method's 21.55%. Furthermore, when context was provided, DoLa achieved an F1 score of 17.94%, compared to our method's 21.55%.

**HQA** On the HQA dataset, fully fine-tuned LLaMa-2-7B models demonstrated superior performance, achieving F1 scores of 41.89% and 41.68% for the LLaMa-2-7B-4k and LLaMa-2-7B-32k models, respectively. In comparison, while the DoLa method recorded an F1 score of 20.42%, our method delivered a modestly superior F1 score of 22.35%.

**Quality** For this dataset, our method significantly outperformed all others. Utilizing JS-divergence as the metric, our method achieved an accuracy score of 53.93%. In contrast, DoLa without context yielded an accuracy score of 33.37%, which is 20.56% lower than that of our method. Even when context was provided,

---

[1] https://huggingface.co/docs/peft/en/package_reference/ia3

DoLa's accuracy score reached only 39.84%, still trailing our method by 14.09%. Across this dataset, our method consistently delivered superior performance. Notably, our method with both kinds of probability distance achieved identical scores.

| Setup | Ctx Size | $\epsilon$ | QAS | QM | NQA | HQA | QuA | Avg. |
|---|---|---|---|---|---|---|---|---|
| **LLaMa-2-7B** | | | | | | | | |
| 4k w.o. Context | 4k | 1x | 7.68 | 12.73 | 10.85 | 22.22 | - | 13.37 |
| 32k w.o. Context | 32k | 1x | 6.30 | 12.79 | 10.61 | 20.03 | - | 12.43 |
| 4k w. Context | 4k | 1x | 16.67 | 14.62 | 14.42 | 32.47 | - | 19.55 |
| 32k w. Context | 32k | 1x | 21.72 | 14.58 | 16.76 | 31.58 | - | 21.16 |
| **LLaMa-2-7B with Finetuning** | | | | | | | | |
| LLaMa-2-7B-4k | 4k | 1.6x | 17.80 | 15.49 | 21.41 | 41.89 | - | 24.15 |
| LLaMa-2-7B-32k | 32k | 12.8x | 29.71 | 16.36 | 28.72 | 41.68 | - | 29.12 |
| **LLaMa-2-7B with Retrieval** | | | | | | | | |
| LLaMa-2-7B-4k w. Retrieval | 4k | 1.6x | 18.29 | 14.33 | 22.28 | 27.95 | - | 20.71 |
| LLaMa-2-7B-32k w. Retrieval | 32k | 12.8x | 24.92 | 15.40 | 19.32 | 22.32 | - | 20.49 |
| **LLaMa-2-7B with DoLa** | | | | | | | | |
| 4k w.o. Context | 4k | 1x | 14.49 | 12.26 | 12.80 | 20.42 | 33.37 | 18.67 |
| 4k w. Context | 4k | 1x | 17.42 | 14.85 | 17.94 | 32.38 | 39.84 | 24.49 |
| **LLaMa-2-7B with our method** | | | | | | | | |
| 4k Ours (non adaptive) | 128k | 30x | 31.43 | 17.21 | 21.55 | 22.35 | 53.93 | 29.29 |
| 4k Ours adaptive($\tau = 0.1$) | 128k | 30x | 31.38 | 17.30 | 21.52 | 14.08 | 53.93 | 27.64 |
| 4k Ours adaptive($\tau = 1$) | 128k | 30x | 31.37 | 17.26 | 21.53 | 14.08 | 53.93 | 27.63 |
| 4k Ours adaptive($\tau = 10$) | 128k | 30x | 31.17 | 17.22 | 21.76 | 14.18 | 53.93 | 27.65 |

Table 1: Experimental outcomes are presented where $\epsilon$ represents the compression ratio. For the LLaMa-2-7B-4k/32k configurations employing retrieval mechanisms, the compression ratio is calculated by dividing the model's context window capacity (4k/32k tokens) by the length of the passages retrieved, consistently set at 2560 tokens. $\tau$ represents the threshold for adaptive decoding.

**QM** In this dataset, variations in results among different methods are minimal. Nonetheless, our approach outperformed all the baselines, achieving a geometric mean ROUGE score of 17.21%, compared to the highest baseline score of 16.36% attained by LLaMa-2-7B-32k with fine-tuning. Although this baseline method requires significantly more time and computational resources, its performance remains inferior to ours. Specifically, DoLa without context recorded a geometric mean ROUGE score of 12.26%, which is 4.95% lower than that of our method. Even when context is provided, DoLa's performance is surpassed by our method, which achieved a geometric mean ROUGE score of 17.21%, exceeding DoLa's 14.08% by 3.13%.

**Overall Performance** Across various datasets, our method demonstrated superior or comparable efficacy. Particularly, compared to DoLa without context or external knowledge, our method significantly outperformed it, with an average performance score of 29.29% against DoLa's 18.67%. Even when DoLa is supplemented with context, our method maintained a performance advantage, achieving an average score of up to 29.29% compared to DoLa's 22.32%. Additionally, our method offers flexibility through adaptive decoding thresholds ($\tau = 0.1$, $\tau = 1$, $\tau = 10$), which slightly adjust the average performance scores to 27.64%, 27.63%, and 27.65% respectively, while maintaining high efficiency and resource utilization. Since the non-adaptive setting achieves the best performance, we adopt it as the default configuration in the subsequent analysis.

# 5 ANALYSIS

## 5.1 PREMATURE LAYER SELECTION STRATEGY

For this part, we adopted a static layer selection variant of DoLa Chuang et al. (2024), where a single layer is fixed as the premature layer and contrastive decoding is conducted by contrasting the probability distribution of next token predicted by the last and the selected premature layers. The selected premature layer ranges from layer 0 to 30, and the results are shown in Figure 2

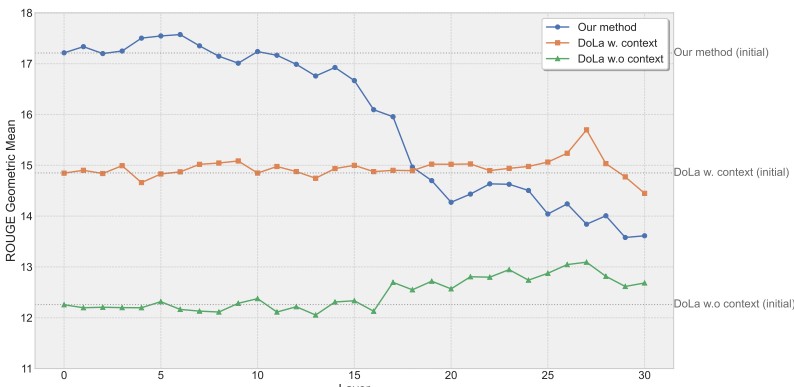

Figure 2: LLaMA-2-7B on QMsum data set with DoLa w.o. context, DoLa w. context and Our method using different premature layers.

From Figure 2, it is clear that our method outperformed DoLa with and without context when fixing an early layer as the premature layer. In particular, DoLa without context attained a low score of 12.26% when fixing the first layer (i.e., layer 0) as the premature layer. Its performance improved when fixing a later layer as the premature layer, but the results are still worse than DoLa with context, which are 12.68% against 14.44% if layer 30 is fixed as the premature layer. DoLa with context had obviously better performance, but it was outperformed by our method. When fixing a layer closer to the last layer as the premature layer, our method's performance decreased. However, it is noteworthy that this part only served to compare the effect of fixed layer selection on the performance of different methods, whereas our method uses dynamically layer selection. Overall, the dot line shows that our method performed the best when fixing the premature layer in the range of layer 0 to 14.

## 5.2 LATENCY & THROUGHPUT

We compared our method to DoLa with and without context, in terms of decoding latency and throughput, using the Quality dataset. As shown in Table 2, DoLa w.o. context had the lowest latency, at 29.0 ms per token, while our method incurred a slightly longer latency, at 35.8 ms per token. DoLa w. context caused the longest latency, of 55.4 ms per token, as the long context increased the reliability but meanwhile increased the latency. By contrast, with our method, the context is greatly compressed and thus the latency brought is relatively low. Regarding throughput (i.e., the number of tokens generated per second), DoLa w. context attained the lowest throughput, at 18.03 tokens per second, which was about half of that by DoLa w.o. context, at 34.48 tokens per second. The throughput of our method was comparable to DoLa w.o. context, at 27.90 tokens per second.

| Model | Latency (ms/token) | Throughput (token/s) |
|---|---|---|
| **DoLa** | 29.0 | 34.48 |
| **DoLa w. context** | 55.4 | 18.03 |
| **Ours** | 35.8 | 27.90 |

Table 2: Decoding Performance Comparison

## 5.3 QUALITATIVE STUDY

Table 5 shows examples generated from the baselines and our method. As we can see, our method predicted the correct answers for each question, but DoLa w.o. context and DoLa w. context failed sometimes. Specifically, for **Q1**, DoLa w. context output relevant information, but it is not as informative as the ground truth due to the limited context window size and thus truncated context. By contrast, the answer produced by our method matches the ground truth. Another example is **Q3**, where our method successfully output "20 evaluators", matching the ground truth, while DeLo with and without context both failed to provide the answer. Overall, the qualitative study shows that our method is reliable to predict correct answers with the context provided. More qualitative examples can be found in the Appendix.

## 6 CONCLUSION AND LIMITATIONS

In this research, we have formulated a novel methodology that improves the reliability of large language model outputs by employing context compression to incorporate extended contexts into prompts, thereby mitigating the issue of hallucinations that arise from the reliance on potentially incorrect or outdated training data. This approach enhances the conventional contrastive learning method by contrasting the final layer of the model with earlier layers to refine the prediction of subsequent tokens. Our method effectively addresses the limitations of existing techniques by enabling the inclusion of extended contextual information, which provides additional cues for question answering tasks. Comprehensive experimental evaluations demonstrate that our method surpasses existing baselines across a variety of datasets, and qualitative assessments confirm the enhanced reliability of our outputs compared to those of the baselines. Moreover, our method exhibits latency and memory consumption comparable to that of the DoLa method. However, unlike modifications to the DoLa approach that merely append context and consequently increase computational demands, our method integrates extended context more efficiently, avoiding excessive computational overhead.

Meanwhile, our study is subject to several limitations. Firstly, due to constraints in computational resources, we have confined our testing to the Llama-2-7B model. Expanding our evaluation to include a broader range of models could further substantiate the effectiveness of our method across diverse architectures and scales. However, extending our method to other models should be relatively seamless, given that it is predicated on the DoLa framework, which has been applied to various models previously. Secondly, although our method demonstrates enhanced performance in question-answering tasks, there are some tasks where its efficacy could still be improved. This may be attributed to potential misalignments between the embedding spaces of the LLMs and the summary embeddings. Thirdly, in terms of context compression, our use of the AutoCompressor model was dictated by the limited resources available. Investigating additional compression methods could facilitate a more thorough exploration and potentially yield more robust findings.

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

# A APPENDIX / SUPPLEMENTAL MATERIAL

## A.1 MORE DETAILS ON DATASETS

In this study, we employ five datasets covering both question answering task and summarization tasks:

- **Quality** (Pang et al., 2021) is a renowned dataset utilized for question-answering tasks. Each entry within this dataset comprises extended contexts accompanied by a question and multiple answer choices. Statistically, the dataset contains 150 articles, with each article averaging 5000 tokens. In total, there are 6737 questions across the dataset.

- **Qasper** (Dasigi et al., 2021) represents another dataset specifically crafted for question-answering tasks. As detailed in (Dasigi et al., 2021), this dataset was compiled from the Semantic Scholar Open Research Corpus. It was chosen for evaluation based on its merits, notably the diversity of question types it encompasses, which range from detailed explanatory answers to straightforward binary yes/no queries.

- **NarrativeQA** (Kočiský et al., 2018) represents a distinctive dataset formulated for question-answering tasks. Unlike previous datasets, NarrativeQA draws its content from complete texts of books sourced from Project Gutenberg and movie scripts from multiple origins. The challenge posed by this dataset involves synthesizing concise answers from the extensive and sometimes unstructured texts of books or movie transcripts.

- **HotpotQA** (Yang et al., 2018) is a well-regarded dataset for the question-answering domain, derived from Wikipedia. The unique aspect of this dataset is its requirement for multi-hop reasoning across several documents to ascertain answers, making it a rigorous test of comprehension and analytical skills. This dataset was chosen due to the diversity of its questions, which span various domains of knowledge.

- **QMSum** (Zhong et al., 2021b) is specifically designed for summarization tasks and comprises transcripts from meetings held in various sectors, including academia and industry. This dataset focuses on query-based summarization, requiring participants during the data compilation phase to condense original dialogue transcripts according to specific queries.

## A.2 MEMORY OVERHEAD

In this section, we evaluate the GPU memory overhead using specific metrics(Chuang et al., 2024), namely: $(a)$ the GPU memory utilization prior to the initial forward pass and $(b)$ the peak GPU memory usage during forward passes. We calculate the memory overhead as the difference $(b) - (a)$, and express it as a percentage of the baseline memory usage, $\frac{[(b)-(a)]}{(a)} \times 100\%$. The findings are presented in Table 3.

| Metric | DoLa | DoLa w. context | Our method |
|---|---|---|---|
| $(a)$ GPU Memory Before Forward (MB) | 12962.0 | 12916.1 | 12791.1 |
| $(b)$ Peak GPU Memory During Forward (MB) | 13421.1 | 20079.5 | 13796.1 |
| $(b) - (a)$ GPU Memory Overhead (MB) | 459.1 | 7163.4 | 1005.0 |
| $\frac{[(b)-(a)]}{(a)}$ GPU Memory Overhead (%) | 3.5% | 55.5% | 7.9% |

Table 3: Memory overhead of inference for LLaMA-2-7B model with various configurations.

The results indicate that our method incurs a memory overhead comparable to that of the DoLa method when no context is provided. For instance, the proportion of GPU memory overhead for DoLa without context is

3.5%, while our corresponding figure is 7.9%. In contrast, DoLa with context exhibits significantly higher memory usage, at 55.5%. This disparity underscores the efficiency of our method, which benefits from employing a context compressor. This compressor reduces the length of the context initially, resulting in more compact embeddings, thereby minimizing the computational load during the attention computation processes.

## A.3 INFERENCE DETAILS

All experiments were conducted using a machine equipped with a single NVIDIA A100 GPU. For experimental execution, the Huggingface Transformers library, version 4.28.1, which has been customized according to (Chuang et al., 2024), was utilized[2]. In terms of decoding strategies, both the DoLa method and our approach employed greedy decoding. The models were operated in evaluation mode with settings adjusted to 16-bit floating-point precision and a batch size of 1.

In the latency and throughput analysis detailed in Section 5.2, we selected 10 examples from the QMsum dataset. During each inference, we recorded the number of tokens generated, aggregating these figures to compute the average value.

For various datasets, different candidate layers were chosen to derive results, as presented in Table 1. The selection of these candidate layers was partially based on guidelines from (Chuang et al., 2024), with further details available in Table 4.

Table 4: Candidate layers for different datasets

| Dataset | Method | Layer Range |
|---|---|---|
| Qasper | DoLa | [16, 18, 20, 22, 24, 26, 28, 30] |
| | DoLa w. context | [16, 18, 20, 22, 24, 26, 28, 30] |
| | Our method | [0, 2, 4, 6, 8, 10, 12, 14] |
| Hotpot_QA | DoLa | [16, 18, 20, 22, 24, 26, 28, 30] |
| | DoLa w. context | [16, 18, 20, 22, 24, 26, 28, 30] |
| | Our method | [0, 2, 4, 6, 8, 10, 12, 14] |
| Narrative_QA | DoLa | [8, 10, 12, 14, 16, 18, 20, 22] |
| | DoLa w. context | [8, 10, 12, 14, 16, 18, 20, 22] |
| | Our method | [0, 2, 4, 6, 8, 10, 12, 14] |
| Qmsum | DoLa | [0, 2, 4, 6, 8, 10, 12, 14] |
| | DoLa w. context | [0, 2, 4, 6, 8, 10, 12, 14] |
| | Our method | [0, 2, 4, 6, 8, 10, 12, 14] |
| Quality | DoLa | [0, 2, 4, 6, 8, 10, 12, 14] |
| | DoLa w. context | [0, 2, 4, 6, 8, 10, 12, 14] |
| | Our method | [0, 2, 4, 6, 8, 10, 12, 14] |

## A.4 STATIC VS DYNAMIC PREMATURE LAYER SELECTION ON OTHER DATASETS

In Figure 3, we show the additional results of static layer selection to compare the performance of our method and DoLa w. context and w.o context, for LLaMA-2-7B models.

## A.5 MORE EXAMPLES FOR QUALITATIVE STUDY

---
[2]https://github.com/huggingface/transformers

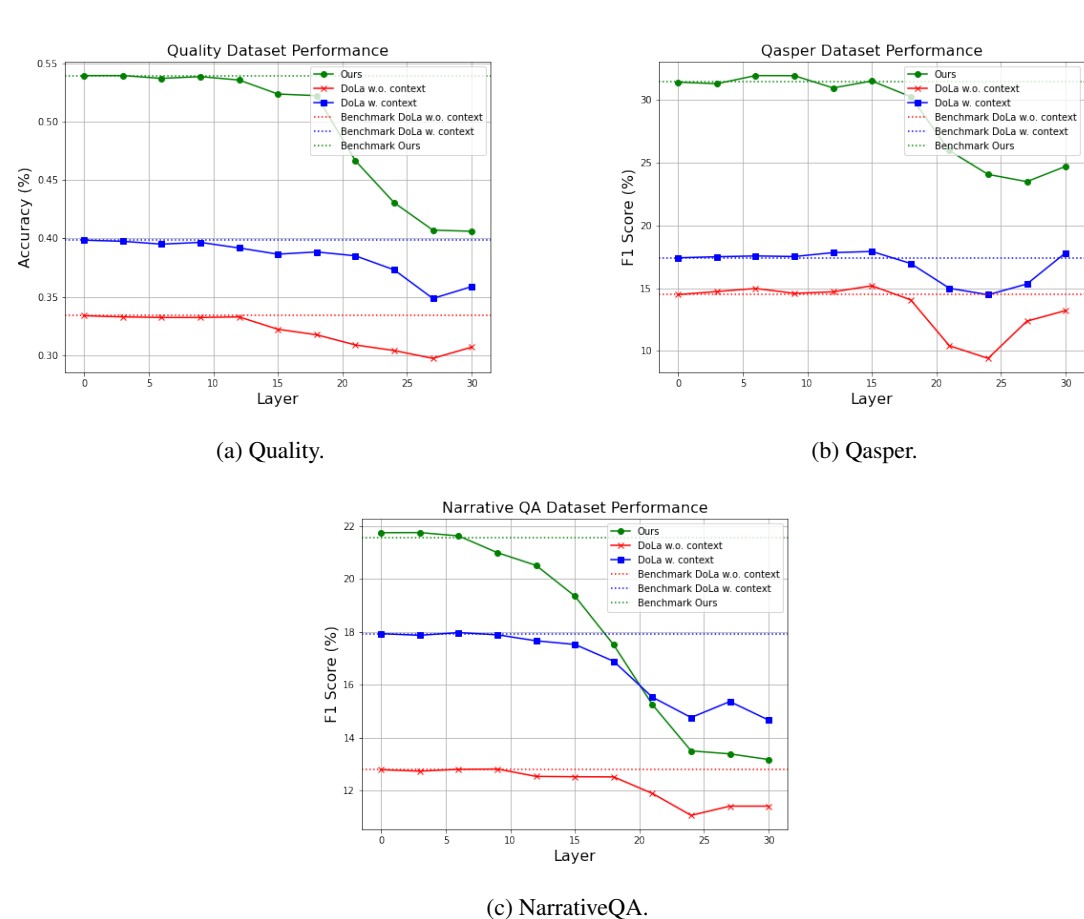

(a) Quality.

(b) Qasper.

(c) NarrativeQA.

Figure 3: Results with static layer selection from different datasets and method.

| Question | **Q1:** *On which benchmarks do they achieve the state of the art?* | **Q2:** *Is the template-based model realistic?* | **Q3:** *Who were the human evaluators used?* |
|---|---|---|---|
| **Context** | **C1:** *Knowledge Base Question Answering (KBQA) systems answer questions by obtaining information from KB tuples BIBREF0, BIBREF1, BIBREF2, BIBREF3, BIBREF4, BIBREF5. For an input question, these systems typically generate a KB ...* | **C2:** *Recently, with the emergence of neural seq2seq models, abstractive summarization methods have seen great performance strides BIBREF0, BIBREF1, BIBREF2. However, complex neural summarization models with thousands of parameters ...* | **C3:** *Recently, with the emergence of neural seq2seq models, abstractive summarization methods have seen great performance strides BIBREF0, BIBREF1, BIBREF2. However, complex neural summarization models with thousands of parameters ...* |
| **Groundtruth** | **G1:** *SimpleQuestions, WebQSP* | **G2:** *Yes* | **G3:** *20 evaluators were recruited from our institution and asked to each perform 20 annotations* |
| **DoLa** | *unanswerable* | *unanswerable* | *unanswerable* |
| **DoLa w. context** | *They achieve the state of the art on single-relation and multi-relation KBQA tasks.* | *unanswerable* | *unanswerable* |
| **Our method** | *SimpleQuestions, WebQSP* | *Yes* | *20 evaluators* |

Table 5: Qualitative study using DoLa w.o. context and w. context, Our method on Qasper Dataset.

| Question | **Q1:** *In what sense does Ro relate to the white young men?* | **Q2:** *Who or what is Leo?* | **Q3:** *What does the Skipper mean by "lady-logic"?* |
|---|---|---|---|
| **Context** | **C1:** *Article: COMING OF THE GODS By CHESTER WHITEHORN Never had Mars seen such men as these, for they came from black space, carrying weird weapons—to fight for a race of which they had never heard.* | **C2:** *Article: CAPTAIN CHAOS By NELSON S. BOND The Callisto-bound Leo needed a cook. What it got was a piping-voiced Jonah who jinxed it straight into Chaos. [Transcriber's Note: This etext was produced from Planet Stories Summer 1942......* | **C3:** *Article: CAPTAIN CHAOS By NELSON S. BOND The Callisto-bound Leo needed a cook. What it got was a piping-voiced Jonah who jinxed it straight into Chaos. [Transcriber's Note: This etext was produced from Planet Stories Summer 1942......* |
| **Groundtruth** | **G1:** *D* | **G2:** *B* | **G3:** *A* |
| **DoLa** | *C* | *C* | *C* |
| **DoLa w. context** | *C* | *D* | *C* |
| **Our method** | *D* | *B* | *A* |

Table 6: Qualitative study using DoLa, DoLa w. context and Our method on Quality Dataset.

