# OpenReview forum: "Can Decoding by Contrasting Layers Really Improve Factuality in Large Language Models?"
_ICLR.cc/2025/Conference — ICLR 2025 Conference Withdrawn Submission_

### Official Review · Reviewer_eDaS · 2024-11-01

**Soundness:** 3
**Presentation:** 3
**Contribution:** 3
**Rating:** 6
**Confidence:** 3

**Summary:**

This paper addresses the challenge of hallucinations in large language models (LLMs), where outputs may diverge from real-world evidence or user inputs. While recent studies have proposed the contrastive decoding strategy DoLa to mitigate output inaccuracy, this approach has limitations due to potential errors from internalized information within LLM parameters. To overcome these issues, the authors propose integrating the contrastive decoding strategy with a long-context encoder, which condenses extensive initial contexts into a more concise format. This method allows for the inclusion of trusted external knowledge within the prompt context despite the constrained window size of LLMs. Extensive experiments demonstrate that this approach significantly enhances the factual accuracy of generated content across various datasets. For example, it improves the performance of LLaMA2-7B models on the Quality dataset by 61.61% compared to the DoLa decoding method, highlighting its effectiveness in improving the reliability of LLMs in generating truthful information.

**Strengths:**

1. The paper presents extensive experimental evaluations, thoroughly demonstrating the effectiveness of the proposed method in enhancing the factual accuracy of LLM outputs across different datasets.
2. The authors provide a meticulous analysis of the proposed approach, offering insights into how the integration of a long-context encoder with contrastive decoding mitigates hallucination issues in LLMs.
3. Interesting findings from the study contribute significantly to the broader research community, offering practical solutions and advancing our understanding of improving the reliability and truthfulness of language model outputs.

**Weaknesses:**

1. While the paper addresses important challenges in LLMs, the proposed method may not represent a groundbreaking innovation within the field, although it effectively builds upon existing strategies. The paper does not adequately distinguish its proposed method from existing contrastive decoding techniques, leaving readers without a clear understanding of the unique contributions and advantages of the new approach.
2. The paper's coverage of related work appears insufficient, failing to provide a comprehensive overview of the state-of-the-art methods and how the proposed approach fits within and advances the current research landscape.
3. The paper's presentation suffers from formatting problems, such as missing formula numbers, which can detract from the overall clarity and professionalism of the work.

**Questions:**

See the above weaknesses.

---

### Official Review · Reviewer_N2Tb · 2024-11-01

**Soundness:** 3
**Presentation:** 3
**Contribution:** 2
**Rating:** 5
**Confidence:** 4

**Summary:**

This paper explores the integration of contrastive decoding with a long-context encoder. Specifically, it employs and fine-tunes the AutoCompressor as a long-context encoder to model external contexts containing essential information. Additionally, it proposes an adaptive decoding strategy that dynamically chooses between predicting the next token based solely on the language model or using contrastive decoding (DoLa) according to uncertainty during decoding, estimated via entropy.

**Strengths:**

1. The introduction of the long-context encoder is effective for compressing the knowledge from long context.
2. The proposed method is more efficient than baselines due to the efficiency of the context encoder.

**Weaknesses:**

1. The motivation behind the proposed adaptive decoding strategy could be more clearly explained, as its rationale and effectiveness might seem questionable. This strategy opts for contrastive decoding only when LM shows uncertainty in predicting the next token, which assumes that contrastive decoding will yield a more accurate prediction in these cases. However, this raises the question: if the authors believe contrastive decoding is generally more effective, why not apply it consistently instead of implementing an adaptive approach? The results of varying the hyperparameter $\tau$ confirm that such an adaptive strategy is not necessary, as always using contrastive decoding reaches the best performance.
2. Considering that the adaptive strategy is not that effective, the novelty of this paper seems limited. It seems primarily to apply AutoCompressor to compress long contexts for LMs, rather than presenting a fundamentally new approach (correct me if I’m wrong).

**Questions:**

1. The content of this paper appears to not address the question posed in the title. I suggest the author revise the title to reflect the content of the paper.
2. Typos:
- Line 286 “in” duplicates.

---

### Official Review · Reviewer_ftpv · 2024-11-03

**Soundness:** 3
**Presentation:** 1
**Contribution:** 3
**Rating:** 3
**Confidence:** 5

**Summary:**

The paper addresses the issue of hallucinations in LLMs. Existing methods like DoLa (contrastive decoding) help reduce hallucinations, but they are limited by relying only on the model's internal parametric knowledge. So the author proposed to use a context encoder that compresses extensive external knowledge and fits within the LLM's context window. An adaptive decoding strategy that dynamically chooses DoLa and normal decoding was adopted. Some performance improvement was shown in the experiments.

**Strengths:**

- The paper proposed a method to combine context encoder with the contrastive decoding DoLa to enhance the factuality of LLMs in long-context scenario.
- The paper addressed the issue of contrasting layerwise parametric knowledge cannot access external knowledge, so it incorporated the context encoder into this method.
- Shows improvements on several tasks and benchmarks

**Weaknesses:**

- The title is very misleading: "Can Decoding by Contrasting Layers Really Improve Factuality in Large Language Models?" --> It seems that you are trying to challenge the idea of DoLa? But in fact, the content of this paper is not like this at all. This paper is extending DoLa to the long-context setting finding DoLa's limitation in long-context, and trying to add some new methods based on / relying on DoLa. In this title, I don't even see the phrase related to "long-context" at all but it's exactly the main topic of this paper. The title is very inappropriate and misleading. It seems just for eye-catching purposes. I cannot accept this paper unless the title is changed. A good title can be something like "Exploring the Limits of Decoding by Contrasting Layers for Long-Context Factuality in Large Language Models"

- Adding a long-context encoder seems to have nothing to do with DoLa at all: DoLa focuses on short-length QA tasks from the beginning. Not sure how this would be a good motivation.

- Figure 1 is also misleading. The DoLa method doesn’t include any context encoder, but the figure shows that the context encoder is part of DoLa, which is not true. You should add a title to the figure such as “Context Encoder + DoLa”

- According to section 3.3, your method will dynamically choose 1) Context Encoder + DoLa 2) Context Encoder + Normal Decoding. If it’s true, then the figure 1 is even more misleading. The caption says the left subfigure is DoLa and the right figure is the proposed method, but the real proposed method actually includes both left and right?

- Some of the math equations or definitions seem redundant, for example, in line 150, the authors defined $V_m$ as the middle layer output but never used it.

- Section 3.2 is PEFT specifically for context compression encoder, the title of the section should better be “Parameter-Efficient Fine-tuning for Context Encoder”

- In Section 3.2, you define the symbol $Q$ as the query vector. However, in Section 3.3, the $Q$ here seems to be the output distribution of early exits, e.g. $Q_L$, $Q_M$, $Q_j$ from $L$, $M$, $j$-th layers, as the query vector cannot compute any JS-divergence. Please define $Q$ properly.

- In Table 1, the row names seem too arbitrary: some include LLaMA-2-7B while some do not; the proposed method seems to have 128K ctx size but the name shows 4k, e.g. 4k Ours (non-adaptive). I am pretty confused about what these row names mean here.

**Questions:**

- I would suggest the authors re-write the paper with the help of those with more experience in academic writing. The idea of this paper is not bad, but the current presentation quality makes it hard for me to accept this paper. The misleading title also made it worse. See Weaknesses.

Typos to fix:
- LLaMa → LLaMA
- LoRa → LoRA

---

### Official Review · Reviewer_JDp3 · 2024-11-03

**Soundness:** 2
**Presentation:** 2
**Contribution:** 2
**Rating:** 5
**Confidence:** 3

**Summary:**

This work proposed a decoding method that improves knowledge grounded language models. The method involves the following steps:

- context compression for reading longer contexts
- finetuning
- adaptive decoding (normal decoding or layer contrastive decoding)

Experiment shows that the proposed method, which compresses 128k token context with a context compression model, achieves the best average performance across four question answering tasks.

**Strengths:**

- The idea of incorporating longer context is well motivated
- The experiment across different benchmarks is convincing and the proposed method achieves a strong avaerage performance.

**Weaknesses:**

In think the major weakness of the paper is that the title, experiment setting, and the conclusion lacks an alignment.

- The title asks such question: "CAN DECODING BY CONTRASTING LAYERS REALLY IMPROVE FACTUALITY IN LARGE LANGUAGE MODELS?"
- The experiments show that simply fine-tuning a llama 32k context model achieves a top performance.
- The conclusion mentioned that the proposed method improved Dola, while Dola under the proposed setting does not outperform simple fine-tuning.

As a result, I believe the reported experiment results neither answered the title question, nor supported the conclusion. However, it might because of the confusing presentation in Table 1 so I failed to get the point. I'll consider adjust the review if authors help me with my confusions in the following question section.

**Questions:**

1. I am curious about the detail of the "llama with fine-tuning" section in table 1. is it full-model tuning or IA$^3$ tuning? Why not showing the 30x compression performance in the fine-tuning section?

2. what does "our method non adaptive" mean in Table 1? is it all normal decoding or all Dola decoding?

---

### Official Review · Reviewer_8YRj · 2024-11-04

**Soundness:** 2
**Presentation:** 3
**Contribution:** 2
**Rating:** 5
**Confidence:** 3

**Summary:**

This paper presents a contrastive decoding method to enhance the factuality of LLM generations by incorporating externel knowledge and adaptive decoding mechanism. Specifically, in order to supplement external knowledge into LLMs, they utilizes context encoder to suppress long-context information. Moreover, this paper also adjust decoding mechanism based on model uncertainty to obtain a more factual response. The authors demonstrate the effectiveness of the proposed method by showing the performance in various dataset and computation cost compared to other conventional contrastive decoding methods.

**Strengths:**

- Well-written and easy to follow the manuscript
- It shows performance improvement on various tasks and better efficiency compared to baselines

**Weaknesses:**

- Limited contribution. The contribution of this work lies in showing that compression with contrastive decoding enhances performances and that adaptive decoding mechanisms may be suboptimal. However, simply combining compression techniques with DoLa lacks sufficient novelty, making the approach more incremental.

- Insufficient analysis. While the study finds adaptive decoding ineffective, it doesn't thoroughly explore why this is the case. Additionally, it lacks analysis of what information context compression actually preserves or whether it introduces unintended side effects(e.g. lossing , . Importantly, it overlooks scenarios where external knowledge might be inaccurate or irrelevant, limiting insights into robustness of the proposed methods under real applications.


- Lack of comparison with other compression methods. The paper does not compare its compression approach with alternative techniques. A broader comparison would clarify the advantages and limitations of this method within the wider context of existing research.

**Questions:**

- Is this method can improve the factuality in long-forma generation tasks?
- Is there evidence supporting the assumption that low uncertainty correlates with accurate answers without the need for contrastive decoding

---

### Note · Authors · 2024-12-08

I have read and agree with the venue's withdrawal policy on behalf of myself and my co-authors.